# Systematic Review: Exosomes as Molecular Messengers in the Development of Obesity-Related Complications in Children

**DOI:** 10.3390/cimb47100865

**Published:** 2025-10-20

**Authors:** Kamila Szeliga, Dominika Krakowczyk, Marcin Chyra, Monika Pietrowska, Tomasz Koszutski, Aneta Monika Gawlik-Starzyk, Lidia Hyla-Klekot

**Affiliations:** 1Upper Silesian Child Health Center in Katowice, The Medical University of Silesia in Katowice, Medykow 16 Street Katowice, 40-752 Katowice, Poland; nika_sm@vp.pl (D.K.); tkoszutski@gczd.katowice.pl (T.K.); agawlik@mp.pl (A.M.G.-S.); lidiahylaklekot@gmail.com (L.H.-K.); 2Department of Pediatric Neurology, Independent Public Healthcare Centre—Municipal Hospital Complex, W. Truchana 7 Street, 41-500 Chorzow, Poland; marcin-chyra@wp.pl; 3Maria Sklodowska-Curie National Research Institute of Oncology, 44-102 Gliwice, Poland; monika.pietrowska@gliwice.nio.gov.pl

**Keywords:** obesity, children, exosomes, complications

## Abstract

Emerging evidence highlights extracellular vesicles (EVs), especially exosomes, as critical molecular messengers linking pediatric obesity to multi-organ complications. This scoping review synthesizes current knowledge on EVs-mediated intercellular communication that exacerbates inflammation, insulin resistance, endothelial dysfunction and organ-specific damage. Data demonstrate that adipose- and endothelial-derived EVs carry bioactive cargo, microRNAs, proteins, and lipids, that modulate key pathways driving metabolic derangements and vascular injury, often preceding detectable clinical biomarkers. Notably, maternal obesity influences EVs composition in breast milk, shaping early-life metabolic programming and offspring risk of obesity. Recent studies underscore the diagnostic and therapeutic potential of EVs in obesity-related conditions such as metabolic-associated fatty liver disease (MAFLD), early renal injury, and cardiovascular dysfunction in children. Furthermore, EVs released in response to exercise or bariatric surgery may mediate systemic metabolic improvements, offering a novel window into personalized interventions. Despite promising findings, standardization of EV isolation and profiling in pediatric research is lacking, and large-scale longitudinal studies are urgently needed. By deepening our understanding of EVs biology, clinicians may advance early detection, risk stratification, and targeted therapies to interrupt the progression from childhood obesity to lifelong metabolic and cardiovascular disease.

## 1. Introduction

Extracellular vesicles (EVs) count as a novel class of circulating biological nanomolecules capable of providing insight into disease patomechanism and processes that remain undetectable by conventional diagnostic instruments such as standard laboratory assays or anthropometric measurements [1]. Various cell types release EVs into the extracellular space and/or blood, which allow EVs to mediate communication even between distant organs and participate in the exacerbation of inflammatory processes and even multi-organ dysfunction. They are involved in various physiological and pathological processes, including immune cell regulation, cell differentiation, infection and cancer [2,3]. The molecular profile of EVs differs due to different composition of proteins, lipids, DNA and microRNA and non-coding RNA and reflects the type and pathophysiological state of the cells from which they originate [1,4].

Pediatric obesity has emerged as a significant global health concern, with its prevalence escalating over recent decades despite dietary interventions, physical activity programs and increased awareness of the obesity epidemic [5]. It has been recognized as a disease and has been included in the International Classification of Diseases with ICD 10: E66. Since 1975, the prevalence of obesity in the world has increased almost threefold, while in 2016, over 25% of the world’s population was diagnosed as overweight or obese (1.9 billion adults, of which 650 million were diagnosed with severe obesity). According to data collected in the sixth round of Childhood Obesity Surveillance Initiative (COSI), which was conducted between 2022 and 2024 in 37 countries and encompassed around 470,000 children, 25% of children aged 7–9 years were living with overweight (including obesity) and in 10% of children obesity was diagnosed. Important differences continue between countries, with the overall prevalence of overweight ranging from 9% to 42% and obesity from 3% to 20%, which confirms that childhood overweight and obesity remain a major public health challenge [5]. The etiology of childhood obesity is multifactorial, involving genetic predisposition, epigenetic programming, environmental influences, unhealthy dietary patterns, and insufficient physical activity. Beyond excess weight, pediatric obesity contributes to early alterations in glucose and lipid metabolism, vascular impairment, and systemic inflammation. Obesity in childhood is a strong predictor of obesity in adulthood, with more than 70% of obese children remaining obese later in life, which significantly increases the risk of more than 200 diseases, including cardiovascular, hepatic, and renal complications as well as mental disorders and cancer (Figure 1).

Preventive measures include family-based lifestyle interventions, structured school programs, policy-driven public health strategies, and early targeted approaches to children at high risk. These strategies emphasize balanced nutrition, regular physical activity, limitation of sedentary behaviors, and psychological support for families [6,7].

In patients with obesity, the excessive accumulation of peripheral subcutaneous adipose tissue (SAT) and interorgan visceral adipose tissue (VAT) plays a key role in disease development. VAT is a hormonally active organ, in which lipid production, storage and metabolism processes continually occur. Hormonal substances produced by VAT, called adipokines/adipocytokines, play a significant role in maintaining the body’s energy regulation by participating in fat and carbohydrate metabolism, vascular remodeling (angiogenesis), blood pressure regulation and immunological processes [8,9]. They act both within adipose tissue (auto- and paracrine action), as well as on distant tissues and organs (classic endocrine action). In obesity, the secretion and action of adipocytokines are dysregulated due to impaired adipogenesis and dysfunction of perivascular adipose tissue [10]. It has been shown that adipocyte cells and adiposal stromal cells are able to secrete extracellular vesicles, especially exosomes which are a novel mode of cell communication for multiple psychological and pathological functions, including proliferation, differentiation, immunomodulation and tumorigenesis. Increased numbers of exosomes released by adipose tissue in patients with obesity have emerged as key mediators in the pathology of organ dysfunction (comorbidities) related to the obesity by direct deregulation of vascular homeostasis [11,12,13,14,15,16,17]. Excessive accumulation of adipose tissue initiates a cascade of pathophysiological processes in which extracellular vesicles play a key regulatory role. Data presented in *Nature* in 2015 [18] show that visceral adipocytes shed exosomal-mediators predicted to regulate key end-organ inflammatory and fibrotic signaling pathways. These processes promote systemic low-grade inflammation, impair glucose homeostasis, elevate blood pressure and contribute to renal impairment associated with obesity, as well as a heightened susceptibility to certain malignancies. Additionally, they are associated with the development of obstructive sleep apnea, mood disorders such as depression, and a progressive decline in overall well-being. Collectively, these alterations significantly increase the risk of cardiovascular complications and premature death [7,19,20]. As such, the study of EVs not only enhances the understanding of obesity-related disease progression but may also facilitate the development of targeted interventions tailored to individual patients.

This review aims to consolidate the most novel knowledge on the role of exosomes in pediatric patients with obesity, focusing on their connection between maternal and child health as well as involvement in metabolic dysregulation and organ-specific complications.

## 2. Materials and Methods

A systematic review of the literature was conducted. The search was performed using electronic databases, including PubMed, Scopus, and Google Scholar, for studies published between 2020 and 2025, taking into account publications prepared according to Minimal Information for Studies of Extracellular Vesicles (“MISEV”) guidelines 2018 and 2013 [21,22].

The keywords used were “extracellular vesicles”, “small vesicles”, “EVs”, “pediatric,” “metabolic diseases,” “lifestyle diseases,” “obesity,” “type 2 diabetes,” and “cardiovascular diseases.” Inclusion criteria were studies focused on pediatric populations (ages 0–18 years) and the use of extracellular vesicles in metabolic assessment. Exclusion criteria included lack of full text in the database, studies on adult populations, animal studies, and those lacking a focus on metabolic assessment through extracellular biology. Data extracted from the selected studies included study design, sample size, disease, identified biomarkers, molecules, EVs and the clinical relevance of the findings. In addition, the references of selected articles were checked with a view to identify papers not detected by our search strategy. At least two authors independently selected articles for inclusion and exclusion criteria. The completed PRISMA checklist is shown in Appendix A.

## 3. Results

Figure 2 presents a schematic overview of the literature selection process. The literature search, including records identified through other sources, identified 629 studies. Following the screening of titles and abstracts, 117 full-text articles were assessed for eligibility and 21 studies met the inclusion criteria. The inclusion and exclusion criteria are described below.

Study selection and data extraction Two independent reviewers (D.K. & K.Sz.) screened all titles and abstracts. Full-text articles were retrieved for potentially eligible studies and assessed for final inclusion. Any disagreements were resolved through discussion or consultation with a third reviewer (L.H.-K.). This ensured methodological rigor and minimized selection bias.

For each included study, the following data were extracted:-Study characteristics (authors, year of publication).-Participant demographics (age, sex, BMI, comorbidities).-EV source and type (e.g., plasma, urine, breast milk, adipose tissue).-Molecular cargo analyzed (miRNAs, proteins, lipids, nucleic acids).-Main outcomes and clinical correlations (insulin resistance, MAFLD, endothelial dysfunction, renal injury, etc.).

Inclusion criteria:-Studies involving participants aged 0–18 years.-Participants classified as overweight or obese according to standardized BMI criteria.-Studies investigating extracellular vesicles (EVs), particularly exosomes, in the context of obesity or obesity-related complications.-Articles assessing molecular cargo of EVs (e.g., microRNAs, proteins, lipids, nucleic acids) and their associations with metabolic, cardiovascular, hepatic or renal outcomes.-Original peer-reviewed research (cohort, case–control, cross-sectional, or interventional designs).-Publications in English between 2020–2025.

Exclusion criteria: -Animal or in vitro studies without validation in pediatric human populations.-Studies not focusing on obesity or obesity-related complications (e.g., general growth, normal-weight controls without an obesity subgroup).-Papers investigating EVs but without specific analysis of exosomal content or functions.-Editorials, conference abstracts, and case reports.

The review followed a systematic search strategy, but the findings were synthesized narratively due to heterogeneity in study designs, EV isolation methods, and outcome measures.

### 3.1. Adipose Tissue and Its Role in Pathophysiology of Obesity-Related Comorbidities

Adipose tissue (AT) is a metabolically active endocrine organ that secretes numerous bioactive peptides known as adipokines, including cytokines, enzymes, and hormones, which regulate energy balance and glucose homeostasis through auto-, para-, and endocrine pathways. In obesity, hypertrophied adipocytes and local hypoxia promote infiltration of pro-inflammatory macrophages (M1 phenotype), creating a chronic low-grade inflammatory state that disrupts lipid and glucose metabolism and drives insulin resistance and endothelial dysfunction [20]. A growing body of evidence shows that extracellular vesicles (EVs), including exosomes, act as key mediators of intercellular communication in these processes. They transport bioactive cargo such as microRNAs (e.g., miR-155, miR-27a, miR-29a), proteins, and lipids that modulate cellular signaling locally and in distant tissues [21]. Recent work by Abd-Elmoniem et al., (2024) [20] demonstrated that both body weight and serum triglyceride levels independently influence circulating EV concentrations and cargo composition in children with obesity [22]. Moreover, EVs from activated or apoptotic endothelial cells are enriched in adhesion molecules, cytokines, and miRNAs that enhance vascular inflammation and leukocyte adhesion [2,24]. AT dysfunction remains a central element linking obesity to metabolic disease. Pro-inflammatory adipokines such as TNF-α and IL-6, together with the recruitment of M1 macrophages, perpetuate inflammation and metabolic stress. Adipocyte-derived exosomes (ADEs) carry cargo that can amplify local inflammation and mediate crosstalk with liver, skeletal muscle, and pancreatic β-cells. For example, exosomal miR-27a from adipocytes inhibits IRS-1 and GLUT4 expression in muscle, while hepatic and muscle-derived EVs modulate adipogenesis, Akt phosphorylation, and insulin sensitivity through miRNAs such as miR-130a-3p and miR-26a [9,21,22,24]. Even gut microbiota-derived EVs can cross the intestinal barrier and impair systemic insulin signaling [25]. Insulin resistance, marked by impaired phosphorylation of Akt (a key serine/threonine kinase), is a hallmark of type 2 diabetes mellitus (T2DM), gestational diabetes mellitus (GDM) and metabolic liver disease. As resistance progresses, compensatory hyperinsulinemia promotes β-cell dysfunction, dyslipidemia, and cardiovascular damage [26,27,28]. Excess adipose storage drives ectopic lipid accumulation in liver and muscle, further aggravating these processes. AT macrophage-derived EVs (ATM-EVs) further exacerbate insulin resistance by modulating key pathways like PI3K-Akt and cytokine release [29,30]. Conversely, EVs from lean or M2-polarized macrophages can improve insulin sensitivity. Recent work by Ying et al., (2021) [24] demonstrated that exosomes derived from M2-polarized bone marrow-derived macrophages (stimulated with IL-4/IL-13) significantly improve insulin sensitivity both in vivo and in vitro. These beneficial effects are mediated by miR-690, an insulin-sensitizing miRNA highly enriched in M2 exosomes, which enhances glucose tolerance and insulin signaling, at least in part by targeting NADK to modulate inflammatory and metabolic pathways. Importantly, the stability and cargo specificity of EVs make them promising diagnostic tools; exosomal miRNAs such as let-7b, miR-144-5p, miR-34a, and urinary PEPCK are being explored as biomarkers of insulin resistance [10,13,31,32]. Current research focuses on engineering EVs as therapeutic vectors for targeted delivery and immunomodulation, particularly in obesity-related metabolic diseases. Understanding the interplay between adipose inflammation, EV-mediated signaling, and metabolic dysfunction will be critical for developing precision strategies to diagnose, prevent, and treat obesity-driven insulin resistance and type 2 diabetes.

Figure 3 illustrates the complex interactions between key organs and adipose tissue via extracellular vesicles. Components include endothelial function, liver signaling through microRNAs, pancreatic involvement in hyperinsulinemia, and effects on skeletal and cardiac muscle. The influence of gut microbiota and systemic insulin resistance is also represented, highlighting the role of extracellular vesicles in these processes.

### 3.2. Endothelial Dysfunction in Obesity and the Role of Exosomes

Endothelial dysfunction in obesity arises from a complex interplay of metabolic, inflammatory, and oxidative stress-related factors. According to the novel knowledge, normal healthy perivascular adipose tissue (PVAT) ensures the dilation of blood vessels through the concerted interplay between following substances: vascular endothelial growth factor (VEGF), TNF-alpha, leptin, adiponectin, insulin-like growth faxctor-1 (IGF-1), interleukin-6 (IL-6), MCP-1, plasminogen activator substance, resistin and angiotensin. In obesity, pathological excessive adipocyte hypertrophy is observed in PVAT, which, via pro-inflammatory macrophage polarization (M1-phenotype), consequently leads to an imbalance of pro- and anti-inflammatory cells and substances. Endothelial dysfunction is a hallmark of obesity-related vascular complications and represents an early step in the pathogenesis of atherosclerosis and hypertension. The chronic low-grade inflammation in obesity disrupts the endocrine and paracrine signaling of adipose tissue, leading to the dysregulation of adipokine secretion, excess of tumor necrosis factor-alpha (TNF-alpha) and 2.5-fold increased vascular expression of endothelin-1 (ET-1) which consequently leads to impaired nitric oxide (NO) bioavailability and enhanced production of reactive oxygen species (ROS) via nicotinamide adenine dinucleotide phosphate (NADPH)-oxidase2 (NOX2) and uncoupling of endothelial nitric oxide synthase (eNOS). Pro-inflammatory adipocyte- and macrophage-derived exosomes contribute to endothelial dysfunction by impairing eNOS expression, increasing oxidative stress, and disrupting intercellular junctions. Specific EV-carried miRNAs have been shown to suppress angiogenic signaling (e.g., VEGF pathways), enhance the expression of adhesion molecules (VCAM-1, ICAM-1), and promote monocyte adhesion and transmigration into the vascular wall. Perivascular adipose tissue (PVAT) becomes dysfunctional in obesity, releasing pro-inflammatory cytokines and oxidative mediators that impair vascular relaxation. Insulin resistance further exacerbates endothelial impairment by blunting insulin-mediated NO production through disrupted PI3K-Akt signaling. Additionally, an imbalance in the L-arginine/asymmetric dimethylarginine (ADMA) ratio impairs endothelial NO synthase (eNOS) activity, contributing to reduced vasodilation [25]. Recent studies have highlighted the role of extracellular vesicles, particularly exosomes, in mediating the molecular crosstalk between dysfunctional adipose tissue and the vascular endothelium. Exosomes derived from hypertrophic or dysfunctional adipocytes carry bioactive cargo, including microRNAs (e.g., miR-155, miR-27a), proteins, and lipids, which modulate endothelial cell function. These vesicles have been shown to downregulate eNOS expression, promote oxidative stress, and disrupt inter-endothelial junction integrity. Moreover, exosomal content from obese individuals may suppress angiogenic pathways and amplify inflammatory signaling cascades within the endothelium. Thus, exosomes represent a novel mechanistic link between metabolic inflammation and vascular dysfunction, offering potential targets for early diagnosis and therapeutic intervention in obesity-related cardiovascular disease. While normal healthy perivascular AT ensures the dilation of blood vessels, obesity-related AT leads to a change in the profile of the released adipocytokines, resulting in a decreased vasorelaxant effect [26,27,28]. The pathology of obesity-related-complications takes place at the level of the vessels, which is manifesting principally as endothelial dysfunction and atherosclerosis [2,25]. Endothelial is a rich source of EVs, and endothelial cell-derived EVs participate in maintaining vascular homeostasis and play a key role in many vascular complications in metabolic diseases. The discovery and characterization of EVs have provided novel insights into the molecular underpinnings of obesity and its long-term sequelae (Figure 2). This shift toward molecular-level interrogation represents a significant advance in pediatric obesity research, enabling the detection of subclinical alterations that precede overt disease and elude standard anthropometric or biochemical assessments. In children and adolescents, who often present with early but nonspecific metabolic perturbations, EV-based molecules hold considerable promise for stratifying individual risk, identifying early organ-specific damage, and tailoring interventions accordingly. By capturing dynamic, tissue-specific signals associated with adipose dysfunction and metabolic stress, EVs-profiling may facilitate a more precise and individualized approach to disease monitoring and therapeutic decision-making in the young obese population [10,29].

### 3.3. Extracellular Vesicles in Maternal—Fetal Communication

The concept of developmental programming posits that the intrauterine environment has lasting effects on offspring phenotype, their neurodevelopment and disease susceptibility [30]. In the context of maternal obesity, EVs emerge as plausible molecular mediators of such cross-generational effects. EVs are abundant in the maternal circulation, amniotic fluid, umbilical cord blood and breast milk—each of which may convey metabolic cues to the developing fetus or neonate. Analyses of EVs in human milk provide a novel enlightenment to better understanding of biologic communication between nursing mothers and infants. Maternal obesity influences the composition of breast milk, including the profile of EVs, which was proven in a study conducted by Shah et al. [31]. They collected the milk samples from a cohort of 30 normal-weight [NW] and 30 overweight [OW] or obese [OB] mothers at 1-month and a subset of 48 of these at 3 months of lactation and concluded that at 1 month of age, the levels of miRNA-148a and miRNA-30b were reduced by 30% and 42%, respectively, in the overweight/obesity group compared to controls. miRNA-148a showed a negative association with infant weight, fat mass, and fat-free mass, whereas miRNA-30b was positively associated with infant weight, percent body fat and fat mass, suggesting a potential role in the infant’s adaptation to enteral nutrition. Cho et al. [32] characterized EVs in breast milk from mothers with obesity and found different expression of 19 microRNAs, including miR-575, miR-630, miR-642a-3p and miR-652-5p, which have been linked to neurological and psychological disorders. Their findings substantiate previous reports about the potential impact of altered breast milk components in mothers with obesity on the neurological development of breastfed infants. According to study conducted by Kunte et al. [33], the adipocyte-derived EVs miRNAs in mothers are potential regulators of fetal adiposity—expression and functionality of miRNAs appear to be influenced by maternal adiposity, hyperglycemia, and micronutrient status during pregnancy. In a novel publication from 2025 written by Diaz et al. [34] on EVs, it was demonstrated that exosomal proteomic signatures at birth differ significantly between infants who were born small (SGA) and appropriate (AGA) for gestational age, despite similar clinical characteristics at enrollment and postnatal catch-up growth. A total of 91 differentially expressed proteins (DEPs) were identified in cord-blood-derived exosomes, with 66 upregulated and 25 downregulated in the SGA subgroup. Hierarchical clustering, PCA, and correlation heatmap analysis confirmed a distinct exosomal proteome profile in SGA infants, suggesting early molecular programming. Notably, two DEPs (up-regulated in SGA), namely PCYOX1 (related to adipogenesis) and HSP90AA1 (related to lipid metabolism and metabolic-dysfunction-associated steatotic liver disease progression) were independent predictors of the hepatic fat fraction at age 7 before the emergence of clinically overt metabolic alterations, such as increased insulin resistance and hepatic fat accumulation. This supports the hypothesis that circulating exosomes may carry early molecular signals associated with developmental programming and long-term metabolic risk. This evidence highlights that miRNAs within human breast milk act as potent immune-regulatory factors, orchestrating immune cell function and the maturation of the infant’s immune system. At the same time, through direct interactions with DNA methyltransferases, they play a pivotal role in the epigenetic regulation of gene expression, with potential long-term effects on health and disease [35]. A deeper understanding of the role of extracellular vesicles (EVs) in human milk and the placenta in the context of early-life programming may open new avenues for preventive strategies targeting obesity risk in offspring. Further investigations are essential to uncover the underlying mechanisms through which maternal EVs modulate infant developmental trajectories and long-term health outcomes.

### 3.4. Extracellular Vesicles in Pathogenesis of Glucose Disturbances, Insulin Resistance and Diabetes

A study by Takaya on adolescents investigated the relationship between extracellular vesicles (EVs) and insulin resistance in adolescents with obesity or type 2 diabetes (T2D). Serum levels of EV markers CD9/CD63 and sonic hedgehog N-terminal (Shh-N) were significantly higher in adolescents with obesity compared to both healthy controls and those with T2D. Serum Shh-N levels were positively correlated with HOMA-IR, β-cell function, insulin, and adiponectin levels. Shh-N emerged as the strongest predictor of insulin resistance, suggesting its potential role as a biomarker of metabolic dysfunction in youth [36]. Findings provided by Elmoniem et al. [20] provided evidence that endothelial dysfunction and early structural vascular changes are already present in young adults with youth-onset type 2 diabetes (Y-T2D) within five years of diagnosis, even in the absence of severe hyperglycemia or dyslipidemia. Using noninvasive MRI, they observed increased coronary artery wall thickness and significantly impaired flow-mediated dilation in both coronary and brachial arteries during physiological stress. These vascular abnormalities were further supported by ex vivo experiments, where plasma-derived small extracellular vesicles (EVs) from Y-T2D patients induced endothelial dysfunction in human coronary artery endothelial cells—marked by reduced phosphorylated eNOS expression, decreased nitric oxide production, elevated oxidative stress, and activation of inflammatory pathways. This was the first study to demonstrate such early vascular alterations in Y-T2D using a combined bench-to-bedside approach. The consistent effects of patient-derived EVs on endothelial cell signaling underscore their potential role as both biomarkers and mediators of vascular injury. These data suggest that EVs may contribute to the pathophysiology of early atherosclerosis in Y-T2D and offer a novel mechanistic link between systemic metabolic dysfunction and cardiovascular risk. Future studies should explore the molecular cargo of these vesicles and their potential as targets for early intervention to prevent long-term cardiovascular complications in this vulnerable population. Kobayashi et al. [37] highlight that relative body weight and serum triglyceride levels were independent predictors of circulating EV concentration, suggesting that EVs may serve as sensitive biomarkers of early metabolic derangement. In their study, proteomic analysis of EVs revealed substantial remodeling of the vesicle cargo in children and adolescents with obesity, with 31 proteins upregulated and 45 downregulated compared to controls. These proteins were implicated in a wide array of biological processes -including protein transport and folding, stress response, leukocyte activation, innate immunity, and platelet degranulation—all of which may contribute to the metabolic, inflammatory, and cardiovascular complications associated with pediatric obesity. Notably, several differentially expressed EVs proteins were linked to neurodevelopmental processes and neuroprotection, indicating that EVs may also reflect obesity-related alterations in central nervous system signaling. The identified EVs were partially derived from adipocytes, hepatocytes, B lymphocytes, and neurons, supporting the concept that circulating EVs represent an integrative signal of tissue-level dysfunction in obesity. Taken together, these results suggest that EVs not only mirror the systemic effects of pediatric obesity but also carry molecular information potentially relevant for early diagnosis, risk stratification, and therapeutic targeting. Further research is warranted to explore the functional significance of these vesicular proteins and their role in mediating long-term complications of childhood obesity.

### 3.5. Extracellular Vesicles and Metabolic-Associated Fatty Liver Disease (MAFLD)

It is worth noting that many proteins and miRNAs can also circulate in free form in plasma. However, EV-associated molecules are more biologically relevant due to their stability, protection from enzymatic degradation, and targeted delivery to recipient cells, which enhances their diagnostic and mechanistic value.

Metabolic-Associated Fatty Liver Disease (MAFLD), as redefined in 2021, is diagnosed when hepatic steatosis involving more than 5% of hepatocytes is identified via imaging modalities such as ultrasonography, magnetic resonance imaging (MRI), or computed tomography (CT). The diagnosis is preferably confirmed by histopathological evaluation (the reference standard) or supported by persistently elevated alanine aminotransferase (ALT) levels—exceeding twice the upper limit of normal for sex—alongside at least one of the following: excess adiposity, prediabetes or type 2 diabetes mellitus (T2DM), or at least two metabolic risk abnormalities (e.g., increased waist circumference, hypertension, hypertriglyceridemia, low HDL cholesterol, impaired fasting glucose, or elevated triglyceride-to-HDL-C ratio) [38]. Epidemiological data suggest that MAFLD affects approximately 3-10% of the general pediatric population, with a marked increase in prevalence to 40-59% among children and adolescents with overweight or obesity, particularly in those with severe obesity. The development and progression of MAFLD are driven by a multifaceted and still not fully understood interplay of genetic, epigenetic, environmental, and metabolic factors. Central to its pathogenesis are insulin resistance and dyslipidemia, which contribute to the buildup of pro-inflammatory mediators. These, in turn, initiate a vicious cycle of lipotoxicity, mitochondrial oxidative stress, and chronic inflammation [39]. MAFLD constitutes a spectrum of liver disease, ranging from simple hepatic steatosis to more advanced stages, including metabolic steatohepatitis (MASH) and cirrhosis. In untreated cases it may progress to eventually hepatocellular carcinoma. Despite growing interest, data regarding the role of extracellular vesicles (EVs) in the pathogenesis of MAFLD, particularly in pediatric populations, remain limited. A recent in vitro study by Baameiro et al. utilized serial ultracentrifugation to isolate EVs from murine adipocytes treated with palmitic or oleic acid, visceral and subcutaneous adipose tissue from obese bariatric patients (“obesesomes”), and steatotic human hepatocytes (“steatosomes”). The results demonstrated that these EVs modulate inflammatory signaling and glucose and lipid metabolism, implicating a mechanistic role in metabolic dysfunction related to obesity and hepatic steatosis [40].

In support of this, a lifestyle intervention study in 18 Latino adolescents with obesity and hepatic steatosis demonstrated a 23% reduction in liver fat following a six-month program. This was accompanied by significant alterations in EV proteomic profiles, including 113 differentially expressed proteins, particularly within the complement cascade pathway. These findings suggest that EV-derived proteins may mediate immunometabolic benefits associated with hepatic fat reduction [41]. Additionally, Zhang et al. performed high-throughput sequencing of exosomal microRNAs (miRNAs) in patients with nonalcoholic fatty liver disease (NAFLD), identifying 2588 miRNAs, of which 80 showed significant differential expression compared to obese controls. Notably, miR-122-5p, miR-27a, and miR-335-5p were highlighted as potentially relevant to NAFLD pathogenesis, and functional analyses linked them to several key biological pathways. These findings support the concept of exosomal miRNAs as candidate biomarkers and molecular targets for diagnosis and therapy [42]. Further, Lischka et al. investigated the association between inflammatory markers derived from adipose tissue macrophages (ATMs) and metabolic comorbidities in 108 children with obesity. Plasma concentrations of IL-1RA, sCD163, and osteopontin (OPN) were significantly elevated in those with MAFLD and metabolic syndrome. IL-1RA and sCD163 correlated strongly with liver fat content, ALT levels, and hepatocyte apoptosis marker CK-18. IL-1RA was also associated with insulin resistance, while sCD163 levels were elevated in participants with impaired glucose metabolism. Since sCD163 can be detected both in soluble form and within EVs, further studies are warranted to dissect the distinct biological roles of these forms in obesity-related inflammation and MAFLD pathophysiology. It is also important to note that many biomolecules, including proteins and miRNAs, may circulate freely in plasma as well as encapsulated within EVs. However, EV-associated cargo is often considered more biologically relevant. First, EVs provide a protective lipid bilayer that shields nucleic acids and proteins from enzymatic degradation in the extracellular environment, thereby markedly increasing their stability and half-life as biomarkers. Second, EV cargo represents a selective and regulated biological packaging process, which suggests a functional rather than random release, enhancing their value as disease-specific indicators. Third, EVs ensure targeted delivery of bioactive molecules into recipient cells through membrane fusion or endocytosis, allowing precise modulation of cellular pathways, whereas soluble proteins and free miRNAs may be more diffusely distributed and less functionally consistent. Collectively, these features support the notion that EV-derived proteins and miRNAs may exert more stable and physiologically meaningful effects on intercellular communication than their circulating counterparts [43].

These findings suggest that ATM-related markers, especially sCD163, may hold promise as non-invasive biomarkers for assessing liver status and metabolic risk in pediatric obesity [44]. Collectively, the available evidence suggests that extracellular vesicles play a multifaceted role in the development and progression of MAFLD. Through the transfer of bioactive molecules, such as miRNAs, proteins, and lipids, EVs mediate cross-talk between adipose tissue and the liver, modulate inflammatory and metabolic pathways, and may reflect early pathological changes in hepatic steatosis. Ongoing and future studies are essential to better define the mechanistic contributions of EVs in pediatric MAFLD and to evaluate their utility as biomarkers and therapeutic targets.

### 3.6. Extracellular Vesicles in Obesity-Related Kidney Disease

Obesity-related glomerulopathy (ORG) is emerging as a significant and underrecognized renal complication in children with obesity, closely linked to progressive chronic kidney disease (CKD). The review by Mangat et al. [45] provides a comprehensive overview of the multifactorial pathophysiology of ORG, underscoring its complex interplay of hemodynamic alterations, adipose tissue–derived inflammation, RAAS overactivation, podocyte injury, insulin resistance, and mitochondrial dysfunction. In particular, podocyte depletion and glomerular hypertrophy are identified as key histopathological features, often presenting clinically as persistent subnephrotic proteinuria in the absence of overt nephrotic syndrome.

Of particular relevance to pediatric care is the observation that the early stages of ORG may remain asymptomatic for extended periods, with renal injury progressing insidiously. The authors emphasize the need for more sensitive biomarkers beyond traditional proteinuria, highlighting emerging candidates such as urinary podocalyxin, KIM-1, NGAL, GluAp, and podocin as promising indicators of early glomerular and tubular damage in children with obesity. Notably, recent studies have suggested that α1-acid glycoprotein (AGP) may outperform albuminuria in detecting early glomerular permeability changes in obese pediatric populations. One study analyzing the miRNA exosomes in urine of children was conducted in 2021 [46], however the authors did not focus on relationship between EVs, obesity and kidney disease in children.

Definite conclusions in pediatric population cannot be drawn; collectively, these insights underscore the urgent need for longitudinal pediatric studies to validate early biomarkers, define standardized diagnostic criteria for ORG in children, and evaluate the long-term renal outcomes of emerging therapeutic strategies, including those targeting inflammatory and metabolic pathways.

### 3.7. Extracellular Vesicles in Cardiovascular Complications in Children with Obesity

Cardiovascular complications represent an increasingly recognized threat in pediatric populations with obesity and related organ-dysfunctions. While overt cardiovascular events remain rare in children, early vascular dysfunction and atherogenesis begin during childhood and adolescence, setting the course for premature cardiovascular disease (CVD) in adulthood [47]. Recent advances highlight exosomes and microparticles as key mediators of intercellular communication in vascular pathophysiology and as emerging biomarkers for early cardiovascular risk [48]. Although most EV studies have been conducted in adults, recent pediatric research provides important insights into their potential as early biomarkers and mediators of vascular complications. In a cross-sectional study, Soltero et al. found that in 271 youth aged 8–20 years, circulating endothelial cells (CEC) counts did not differ across BMI categories. However, severe obesity was associated with significantly greater CEC activation compared with normal weight peers and number correlated with increased body fat percentage and systolic blood pressure, while greater CEC activation was linked to visceral adiposity and elevated non–HDL cholesterol. These findings suggest that CEC activation, rather than absolute counts, may serve as an early marker of endothelial dysfunction and cardiometabolic risk in youth [49]. Jang et al. in their cross-sectional study investigated the association between circulating endothelial microparticles (EMPs) and their activated forms, with adiposity and subclinical cardiovascular disease (CVD) risk in 280 youth aged 8–20 years across various BMI categories. Using flow cytometry and dual X-ray absorptiometry, researchers found that levels of microparticles, EMPs and activated EMPs, increased significantly with higher adiposity. Compared to youth with normal weight, those with overweight/obesity and severe obesity had up to 3.4 times higher microparticle and EMP levels. Elevated microparticles and EMPs were also linked to unfavorable vascular measures. Importantly, higher adiposity was associated with significantly increased odds of elevated activated EMPs. These findings suggest that microparticles, particularly EMPs, may serve as early biomarkers of cardiovascular risk in children with excess adiposity [50]. EVs are emerging as pivotal molecular messengers in the development of cardiovascular complications in pediatric population with obesity. While much remains to be elucidated, early evidence suggests that EVs hold promise both as non-invasive biomarkers of vascular injury and as potential therapeutic targets. Advancing this field will require interdisciplinary collaboration, standardized methodologies, and well-designed pediatric cohorts to fully harness the translational potential of EVs in reducing lifelong cardiovascular risk.

### 3.8. Exercise-Induced Extracellular Vesicles and Metabolic Adaptation in Pediatric Obesity

Physical activity exerts well-documented benefits on metabolic health, with emerging evidence suggesting that a portion of these effects is mediated through extracellular vesicles (EVs). Exercise has been shown to induce the release of EVs enriched in specific microRNAs (miRNAs) that regulate gene expression in distant tissues, modulating key metabolic and inflammatory pathways. In individuals with obesity, exercise-responsive EVs appear to carry miRNA cargo involved in improving insulin sensitivity and lipid metabolism, supporting a mechanistic role of EVs in the systemic benefits of physical activity. In a pivotal study by Pierdona et al. (2022) [51], adolescents with obesity who exhibited favorable metabolic responses to resistance training were found to release EVs with distinct biophysical properties during acute aerobic exercise. These individuals produced significantly larger EVs with greater protein content and a molecular profile suggestive of larger vesicle subtypes (e.g., increased MMP2, decreased TSG101), both at rest and during physical exertion. Notably, EV size positively correlated with improvements in insulin sensitivity, suggesting a potential utility for EV parameters as predictive biomarkers of exercise responsiveness. Further supporting this concept, Sullivan et al. [52] investigated the impact of aerobic and resistance training on skeletal muscle-derived EVs in children with obesity. Their findings revealed that obesity is associated with significant alterations in the miRNA cargo of small EVs, particularly miRNAs targeting inflammatory and anabolic signaling pathways such as Wnt/β-catenin, IGF-1, and PI3K/AKT. Importantly, short-term exercise interventions led to a shift in EV miRNA profiles toward an anti-inflammatory signature, with reduced targeting of IL-6, IL-10, and NF-κB pathways. These molecular changes coincided with a downregulation of pro-inflammatory markers in skeletal muscle, highlighting the role of EVs as mediators of exercise-induced anti-inflammatory. Rigamonti et al. [53] expanded on this knowledge by characterizing circulating EV profiles in response to acute moderate-intensity exercise. Their study demonstrated a general post-exercise reduction in total EVs, particularly microvesicles, with a more robust release observed in normal-weight controls compared to obese participants. Muscle-derived EVs increased, whereas platelet-derived (CD61^+^) EVs decreased, with changes persisting for up to 24 h post-exercise. Notably, levels of total EVs, exosomes, and CD61^+^ EVs correlated with insulin resistance as measured by HOMA-IR. Sex-related differences were also evident, with females exhibiting higher microvesicle but lower exosome concentrations post-exercise than males. Collectively, these studies underscore the complexity and relevance of EV-mediated signaling in the context of exercise and metabolic health. EVs may serve not only as biomarkers of individual response to physical training but also as active players in mediating metabolic improvements, particularly in youth with obesity.

### 3.9. Extracellular Vesicles in the Context of Bariatric Surgery and Metabolic Improvement

Bariatric surgery remains a cornerstone therapeutic intervention for individuals with severe obesity, yielding substantial and sustained weight loss alongside improvements in metabolic parameters. Emerging evidence indicates that this surgical approach induces significant alterations in the profile of circulating extracellular vesicles, which may reflect shifts in adipose tissue functionality and systemic metabolic homeostasis. Notably, EVs derived from visceral and subcutaneous adipose depots of patients undergoing bariatric procedures display distinct proteomic signatures, suggesting depot-specific contributions to metabolic regulation. To date, the majority of studies exploring the impact of bariatric surgery on EVs dynamics have been conducted in adult populations. Data in children and adolescents are scarce. However, preliminary findings suggest that EVs may serve as promising biomarkers of weight-loss-associated changes in insulin sensitivity, inflammation, endothelial health, and renal function in pediatric cohorts with obesity. In a study by Choi et al., urinary EVs RNA profiles were characterized in obese adults before and after bariatric surgery, in comparison to healthy controls. RNA sequencing revealed differential expression of 1343 small RNAs-including transfer RNAs and microRNAs, between obese individuals and controls, with several transcripts demonstrating reversal following significant weight loss. Correlation analyses revealed associations between specific RNA species and clinical parameters such as body weight, lipid profile, glycemic control, kidney function, and albuminuria. These findings support the hypothesis that urinary EVs small RNAs are modulated by obesity and respond dynamically to weight reduction, underscoring their potential as non-invasive biomarkers for obesity-related renal dysfunction [54]. To our knowledge, only one study has thus far addressed this topic in the pediatric population. A U.S.-based investigation involving 58 adolescents with severe obesity evaluated the role of small extracellular vesicles (EVs) following vertical sleeve gastrectomy (VSG). The procedure was associated with significant metabolic improvement and was accompanied by marked changes in EVs RNA cargo. Notably, liver-derived EV RNA content decreased after VSG and correlated with body mass index (BMI), circulating leptin levels, and branched-chain amino acids (BCAAs), suggesting a mechanistic role in systemic metabolic regulation. Furthermore, adipose-derived EVs were associated with changes in insulin resistance and hepatic biomarkers. These data support the existence of a dynamic inter-organ communication axis between liver and adipose tissue mediated by EVs and highlight EV cargo as a potential therapeutic target to mimic the metabolic benefits of bariatric surgery [55].

In Table 1 we summarize current knowledge on the role of extracellular vesicles (EVs), including exosomes and microparticles, in obesity-related comorbidities in pediatric and adolescent populations. EVs carry bioactive molecules (miRNAs, proteins, lipids) that influence metabolic signaling, inflammation, endothelial function, and multi-organ complications.

## 4. Discussion

In this systematic review, we summarized the role of extracellular vesicles (EVs) in obesity-related complications, with a particular focus on pediatric populations. Although much of the literature originates from adult cohorts, the pediatric data included here demonstrate that EVs act not only as molecular messengers but also as active mediators of metabolic and vascular dysfunction. This is highly relevant, since complications of obesity often begin subclinically during childhood, yet most clinical manifestations emerge only later in life. Our analysis revealed that adipose tissue-derived EVs in children transport pro-inflammatory and pro-oxidative molecules that promote endothelial dysfunction, vascular inflammation and insulin resistance. While similar mechanisms have been described in adults, pediatric studies suggest that these processes may occur earlier and may be more strongly influenced by developmental, hormonal and growth-related factors. Importantly, this highlights the limitation of extrapolating adult data directly to children. Endothelial-derived EVs were associated with coagulation imbalance and vascular inflammation in adolescents, which suggests that endothelial EVs may serve as sensitive, early indicators of vascular stress in children, even before clinical symptoms are detectable. A recurrent theme across the included studies was the distinct cargo composition of EVs, encompassing proteins, lipids and microRNAs, that differentiated obese children from their lean peers. This selective packaging process underscores the functional role of EVs in shaping intercellular communication and supports their value as disease-specific biomarkers. The pediatric data, although limited in quantity, emphasize the potential for EVs to serve as early indicators of metabolic dysfunction long before irreversible organ damage occurs. From a translational perspective, EVs offer several promising applications in pediatric obesity:
-Molecular messengers for early detection of complications such as metabolic-associated fatty liver disease (MAFLD), endothelial dysfunction, insulin resistance, type 2 diabetes, kidney dysfunction.-Therapeutic targets, since modulation of EV release or uptake may attenuate inflammation and vascular damage.-Dynamic indicators of intervention efficacy, given that EV signatures have been shown to change following lifestyle modifications, structured exercise programs, or bariatric surgery.

Nevertheless, pediatric data remain scarce and largely descriptive. Future studies must address whether EV profiles in children can reliably predict clinical outcomes and therapeutic responses. In addition, standardized methodologies for EV isolation and characterization in pediatric samples are needed to ensure reproducibility and comparability across studies.

## 5. Clinical Significance and Innovation

Research on small extracellular vesicles is rapidly evolving, with exosomes increasingly recognized as both molecular tissue messengers and therapeutic targets. Their ability to influence vascular homeostasis, inflammation, and metabolic signaling suggests several clinical applications:-Early detection of complications: EV-derived miRNAs and proteins may act as sensitive, non-invasive biomarkers for the early identification of MAFLD, vascular dysfunction, or insulin resistance, before irreversible damage occurs.-Therapeutic targeting: Modulation of exosome biogenesis, release, or uptake represents a novel therapeutic approach. Moreover, exosomes may serve as natural carriers for drugs or RNA-based therapies, enabling targeted delivery with reduced systemic toxicity.-Monitoring interventions: EV profiles respond dynamically to lifestyle modifications, physical activity, and bariatric surgery, making them valuable biomarkers to track treatment efficacy and therapeutic “windows of opportunity.”

## 6. Limitations

We acknowledge several limitations of our review:-Timeframe restriction: Our systematic search covered studies published within the last 5 years. While this was a deliberate decision to capture the most up-to-date and methodologically relevant data in this rapidly evolving field, it may have led to omission of earlier but still valuable publications,-Heterogeneity of methodologies: The included studies used diverse techniques for EVs isolation, characterization, and biomarker profiling. This heterogeneity limits the possibility of direct comparison and prevents formal meta-analysis-Small number of pediatric studies: Compared to adults, studies focusing specifically on extracellular vesicles in children remain scarce. This limits the strength of conclusions directly applicable to the pediatric population and underscores the need for large-scale, longitudinal pediatric studies.-Narrative synthesis: Owing to the diversity of study designs and outcome measures, we performed a narrative rather than quantitative synthesis. Although this allowed us to summarize key findings, it reduces the ability to assess pooled effect sizes,-Potential publication bias: As in most systematic reviews, there is a possibility that negative or null findings remain unpublished, which may lead to an overestimation of positive associations.

## 7. Future Perspectives

Future research should focus on:-Elucidating the longitudinal dynamics of EVs during the progression of pediatric obesity and its treatment.-Developing EV-based risk stratification tools to personalize prevention and therapy.-Advancing EV engineering approaches, e.g., loading exosomes with insulin-sensitizing or anti-inflammatory molecules to enhance therapeutic precision.

By linking cellular-level mechanisms with clinical outcomes, EV research has the potential to shift pediatric obesity care from late detection of complications toward early, EV-based precision prevention and treatment. In summary, EVs represent a promising frontier for risk stratification, early diagnosis, and novel therapeutic strategies in pediatric obesity, with the potential to alter the long-term trajectory from childhood obesity to lifelong metabolic and cardiovascular disease.

## 8. Conclusions

Extracellular vesicles (EVs) have emerged as central molecular mediators in pediatric obesity, linking adipose tissue dysfunction with systemic complications across cardiovascular, hepatic, renal, and endocrine systems. By transporting microRNAs, proteins, and lipids, EVs influence insulin signaling, endothelial health, inflammation, and fibrotic remodeling. Their cargo reflects tissue-specific stress earlier than conventional biomarkers, highlighting their promise as non-invasive tools for early diagnosis and risk stratification. Maternal metabolic status and breast milk–derived miRNAs further shape early-life immune and metabolic programming, suggesting a role in intergenerational risk transmission. However, key gaps remain: large, longitudinal pediatric studies are needed to validate EV-based biomarkers, define normative ranges, and clarify predictive value across development. Future translational work should explore how lifestyle modification, pharmacotherapy, and bariatric surgery alter EVs release and function, offering new opportunities for early intervention and personalized care.

## Figures and Tables

**Figure 1 cimb-47-00865-f001:**
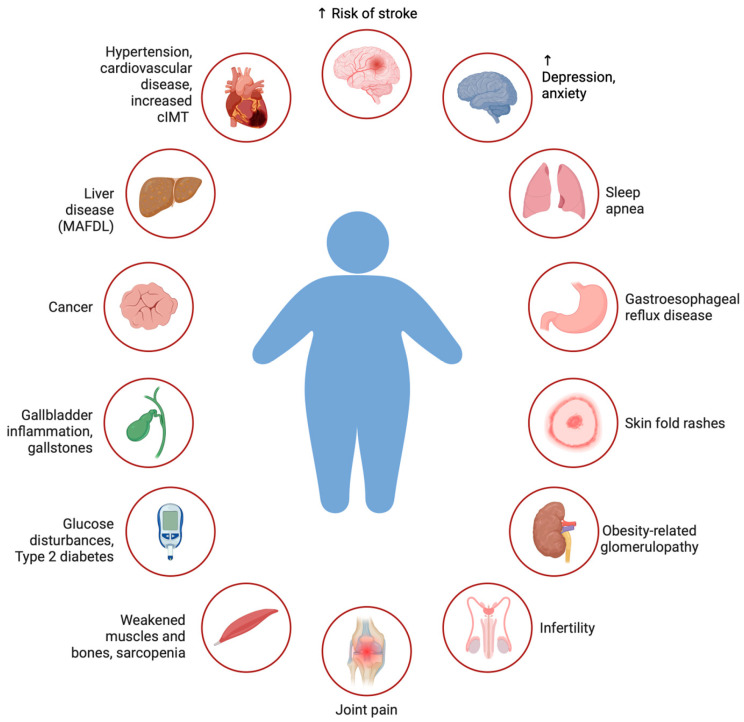
The overview of obesity-related complications.

**Figure 2 cimb-47-00865-f002:**
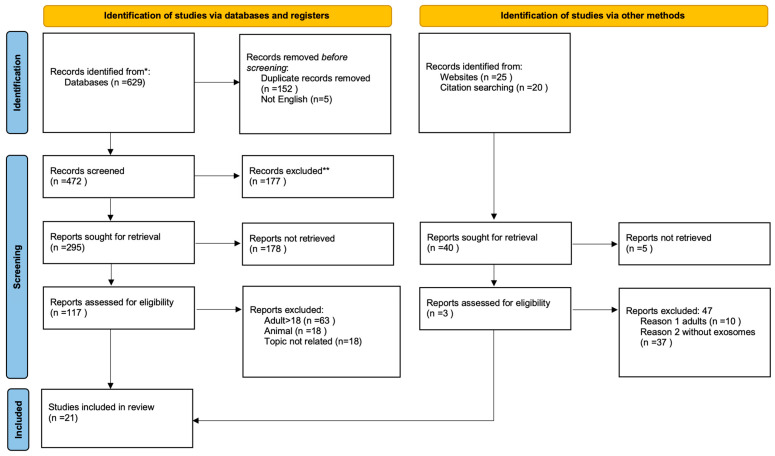
Study selection process PRISMA flowchart [23] (This work is licensed under CC BY 4.0. To view a copy of this license, visit https://creativecommons.org/licenses/by/4.0/). * Consider, if feasible to do so, reporting the number of records identified from each database or register searched (rather than the total number across all databases/registers). ** If automation tools were used, indicate how many records were excluded by a human and how many were excluded by automation tools.

**Figure 3 cimb-47-00865-f003:**
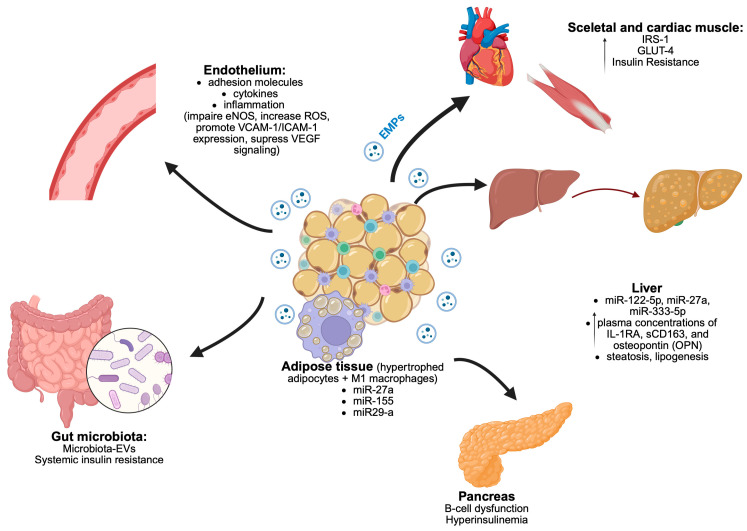
The interaction between adipose tissue and key organs via extracellular vesicles. Legends: eNOS—Endothelial NOS (eNOS), also known as nitric oxide synthase 3 (NOS3), ROS—Reactive Oxygen Species, VCAM-1—vascular cell adhesion molecule 1, ICAM-1—Intercellular Adhesion Molecule 1, VEGF—Vascular Endothelial Growth Factor, IRS-1—Insulin Receptor Substrate-1 protein, GLUT-4—Glucose Transporter 4. Upward arrow means Elevated plasma concentrations of IL-RA, sCD163 and osteopontin (OPN).

**Table 1 cimb-47-00865-t001:** Summary of the role of extracellular vesicles (EVs) in obesity-related comorbidities.

Clinical Context	Role of EVs	Key References
Adipose tissue dysfunction	Adipocyte-derived EVs carry miRNAs (miR-27a, miR-155, miR-29a) affecting insulin signaling, glucose uptake, and inflammation; macrophage-derived EVs modulate PI3K-Akt pathway; M2-derived EVs with miR-690 improve insulin sensitivity.	Kobayashi et al., 2024 [37]; Ying et al., 2021 [24]
Endothelial dysfunction	Adipocyte- and macrophage-derived EVs impair eNOS, increase ROS, promote VCAM-1/ICAM-1 expression; suppress VEGF signaling; link adipose inflammation with vascular dysfunction.	Engin 2017 [25]; Fang et al., 2024 [2]
Maternal-fetal communication	Breast milk and placental EVs convey miRNAs (miR-148a, miR-30b, miR-575, miR-630, miR-642a-3p) affecting infant adiposity, neurodevelopment, and immune maturation; cord blood EV proteome predicts later hepatic fat fraction.	Shah et al., 2021 [31]; Cho et al., 2022 [54]; Kunte et al., 2023 [33]; Díaz et al., 2025 [34]
Glucose disturbances/T2DM	Circulating EVs elevated in adolescents with obesity/T2D; markers (CD9/CD63, Shh-N) predict insulin resistance; patient-derived EVs induce endothelial dysfunction; EV proteome remodeling reflects metabolic derangement.	Takaya 2023 [36]; Abd-Elmoniem et al., 2024 [20]; Kobayashi et al., 2024 [37]
MAFLD (Metabolic-Associated Fatty Liver Disease)	Adipocyte/hepatocyte-derived EVs (‘obesesomes’ and ‘steatosomes’) modulate inflammation and lipid metabolism; exosomal miRNAs (miR-122-5p, miR-27a, miR-335-5p) linked to MAFLD progression; EV proteins (sCD163, IL-1RA) correlate with liver fat and ALT.	Lago-Baameiro et al., 2025 [40]; Zhang et al., 2022 [42]; Lischka et al., 2023 [44]; DiStefano et al., 2024 [41]
Kidney disease: Obesity-Related Glomerulopathy (ORG)	Urinary EV miRNAs proposed as early biomarkers; AGP and podocyte injury markers potentially superior to albuminuria; need for pediatric validation studies.	Mangat et al., 2023 [45]; Levin-Schwartz et al., 2021 [46]
Cardiovascular complications	Endothelial-derived EVs, circulating endothelial cells (CECs), and microparticles (EMPs) reflect early vascular injury. Severe obesity is linked with greater CEC activation, while higher CEC number correlates with adiposity and blood pressure. EMPs and activated EMPs increase with adiposity and are associated with arterial stiffness and subclinical atherosclerosis.	Soltero et al.; 2021 [49]; Jang et al., 2022 [50]
Exercise-induced adaptations	Exercise-induced EVs enriched in specific miRNAs regulating insulin sensitivity, inflammation, Wnt/IGF-1/PI3K pathways; distinct EV size/protein content predicts exercise responsiveness.	Pierdona et al., 2022 [51]; Sullivan et al., 2022 [52]; Rigamonti et al., 2022 [53]
Bariatric surgery & metabolic improvement	Bariatric surgery alters EV cargo (RNAs, proteins) reflecting improved metabolic state; liver/adipose EVs correlate with insulin resistance, hepatic biomarkers, leptin, BCAA levels; urinary EV RNAs proposed as biomarkers of weight-loss benefits.	Choi et al., 2022 [54]; Kim et al., 2024 [55]

Legend: EVs = extracellular vesicles; EMPs = endothelial-derived microparticles; MAFLD = Metabolic-Associated Fatty Liver Disease; ORG = obesity-related glomerulopathy.

## Data Availability

No new data were created or analyzed in this study. Data sharing is not applicable.

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
