# Peer review of "Systematic Review: Exosomes as Molecular Messengers in the Development of Obesity-Related Complications in Children"

_cimb, 2025, doi:10.3390/cimb47100865_

Round 1

Reviewer 1 Report

Comments and Suggestions for Authors

In the manuscript, entitled »Systematic Review: Exosomes as Molecular Messengers in the Development of Obesity-Related Complications in Children« submitted to CIMB for a potential publication, the authors present their systematic review investigating the role of exosomes in development of childhood obesity complications. The article is interesting and well-written, anyway, there are some parts to be discussed and improved before reconsidering it again. Therefore, I am of opinion that in the present form it is not good enough to be published and needs some revision.

My comments:

  1. In Introduction, the aetiology of obesity in children should be briefly described as well as its complications and tracking from childhood to adulthood. In addition, some preventive measures with possible targeted interventions should be included.
  2. In Introduction, the interconnection of obesity with cardiovascular health as well as renal and liver pathology should be delineated.
  3. The process of article search should be presented in more detail, e.g. how were the articles selected, with how many researchers etc.
  4. One or two figures, presenting the field of extracellular vesicles (EVs) and exosomes, would be of value. This way the article will be much more readable, especially for those who are not exactly from this field as well as not so technical. In addition, a graphic abstract should be included.
  5. The sentence that starts in line 143 and ends in line 144 has no citation, which is a problem in itself and also from a semantic point of view, because there is not stated which cargo these EVs carry with positive correlation and this is important/interesting information.
  6. In line 350 the article briefly mentions the fact that some proteins are also soluble in blood and not only in EVs, but miRNAs can also be "free". A short discussion should be done, commenting  why these are more biologically relevant or interesting in EVs.
  7. What could be the clinical significance of EVs, e.g. what the options are to become therapeutic targets? It is mentioned several times throughout the text, anyway, no options are shown.
  8. Some future perspectives about the research in the field should be delineated.
  9. A stylistic correction in line 259 to change the sentence because it doesn't make sense right now.

Author Response

We would like to express our sincere gratitude to the Reviewer for the thorough evaluation, constructive feedback, and insightful suggestions, which have greatly contributed to improving the quality, clarity, and scientific value of our manuscript. Below, we provide a detailed, point-by-point response to each of the Reviewer’s comments. All corresponding modifications have been incorporated into the revised version of the manuscript, with the changes highlighted in yellow for ease of reference.

1 In Introduction, the aetiology of obesity in children should be briefly described as well as its complications and tracking from childhood to adulthood. In addition, some preventive measures with possible targeted interventions should be included.

2: In Introduction, the interconnection of obesity with cardiovascular health as well as renal and liver pathology should be delineated.

Ad 1 and 2 : We have expanded the Introduction to include a concise overview of the etiology of childhood obesity, its major complications, and the strong tracking of obesity into adulthood. Furthermore, we added a short section on preventive strategies and targeted interventions (e.g., early lifestyle modifications, family-based approaches, and public health strategies) as well as to to highlight the link between childhood obesity and cardiovascular, renal, and liver complications, which are further elaborated in subsequent sections of the manuscript.
Changes introduced in Introduction, page 2, lines 56-66

3: The process of article search should be presented in more detail, e.g. how were the articles selected, with how many researchers etc.

Ad 3. We have added more detail to the Materials and Methods section, describing the selection process, the number of reviewers involved (at least two researchers independently screened titles/abstracts and full texts), and how disagreements were resolved.

4: One or two figures, presenting the field of extracellular vesicles (EVs) and exosomes, would be of value. In addition, a graphic abstract should be included.

Response: It would be a great pleasure to add more detailed pictures and figures.  
To Editor and reviewer: Could you please clarify what are the specific requirements for graphical abstracts in Current Issues in Molecular Biology(CIMB)?

In particular, we would like to know:

  • the recommended file format (e.g., TIFF, PNG, JPEG, PPTX, PDF),
  • the resolution requirements (e.g., 300 dpi),
  • the dimensions (e.g., width × height in pixels or cm),
  • whether the abstract should be submitted separately or embedded in the main manuscript file,
  • and any style preferences (e.g., minimal text, color schemes, font guidelines).

This information will help us prepare a figure that meets the journal’s standards

  1. The sentence in line 143–144 has no citation, and it is unclear which cargo is carried with positive correlation.

Ad 5: We have revised this sentence for clarity and added appropriate citation indicating the types of EV cargo. Lines 190-197.

6: In line 350, the article briefly mentions that some proteins are soluble in blood and not only in EVs, but miRNAs can also be "free". A short discussion should be included on why these are more biologically relevant in EVs.

Response: We have added a short discussion explaining why EV-associated molecules are more biologically relevant compared to their soluble/free counterparts.

7: What could be the clinical significance of EVs, e.g. therapeutic targets?

8: Some future perspectives about the research in the field should be delineated.

Ad. 7 and 8:  We thank the reviewer for this valuable suggestion. We have expanded the Discussion to emphasize the potential clinical significance of EVs, including their role as biomarkers for early detection of complications (e.g., MAFLD, endothelial dysfunction), as therapeutic vehicles for targeted delivery, and as dynamic indicators of treatment response. Furthermore, we have added a section on future perspectives, highlighting the need for longitudinal pediatric studies, EV-based risk stratification tools, and engineered EVs as novel therapeutic strategies. 

9: A stylistic correction in line 259.

Ad. 9 The sentence was corrected for clarity and style.
Lines: 312 - 314

Reviewer 2 Report

Comments and Suggestions for Authors

The manuscript by Kamila Szeliga and co-authors is devoted to the role of exosomes in obesity-related diseases. The review is focused on children, which is a relatively rare view to the problem.

1. Unfortunately, the Introduction is written rather formally, and the focus of the review disappears while reading. While slight corrections of the first and second paragraphs could be enough to make the text more fresh and interesting, the third paragraph should remain within the children-oriented discussion. The references should be used accordingly and with care.

2. The "Methods" section does not provide enough information. One of the main features of a scientific paper is its reproducibility by independent researchers. Thus, the description should enable such principle. For example, some keywords are listed in the section, but this is not enough. While testing the approach one can initiate a search in PubMed. The search for

((((((((extracellular vesicles[Title/Abstract]) OR (small vesicles[Title/Abstract])) OR (EVs[Title/Abstract])) OR (pediatric[Title/Abstract])) OR (metabolic diseases[Title/Abstract])) OR (lifestyle diseases[Title/Abstract])) OR (obesity[Title/Abstract])) OR (type 2 diabetes[Title/Abstract])) OR (cardiovascular diseases[Title/Abstract]) within 5 years provides 15,218 results. The description of "Methods" should be modified, so anyone could obtain almost the same search results as you did.
3. In addition, the distribution of publications by years implies a 10-year period should be covered, instead of just 5-years. A substantial amount of papers is lost with the search parameters used in the current version.
4. The Results section has mostly the same flaw as the Introduction. Only the last lines (199-208) of the section mention children, while no such context is obvious from the other parts of the Results. The text should be reviewed according to the title of the paper. The difference between children and adults must be reviewed too.

5. The manuscript lacks figures and tables, although two figures are mentioned, and one of them is indeed included as Supplementary. Comparison to other Open Access Systematic Reviews should enable improvement of the manuscript, its content and representation.
6. Finally, the choice of topics for the Discussion is not clear from the text. A good discussion always evidently follows the results and uses other literature for a better comparison.

Author Response

We are deeply grateful to the Reviewer for the careful reading of our manuscript, as well as for the insightful and constructive comments. We are convinced that these valuable suggestions have substantially improved both the clarity and scientific rigor of our work. Please find our detailed responses below.

Ad 1.
We thank the Reviewer for this important observation. The Introduction has been carefully revised to improve readability and maintain a clear, child-centered focus throughout. The paragraphs were refined to provide a more engaging narrative linking EVs to obesity; hopefully now explicitly emphasizes the pediatric perspective. References were revised to ensure that citations are balanced and appropriately support the child-centered scope.

Ad 2.
We fully agree with the Reviewer that applying the broader set of filters and keywords, as suggested, indeed generates a substantially higher number of results. However, the vast majority of these publications do not correspond to the specific scope of our review, namely the intersection of extracellular vesicles and pediatric obesity. In order to achieve the most accurate and comprehensive assessment of all potentially relevant studies, we opted for a meticulous manual screening process. This was admittedly a lengthy and demanding task, but it allowed us to carefully evaluate every title for relevance. The initial filtering step yielded 627 publications, which were subsequently reviewed in detail according to our predefined inclusion and exclusion criteria. We deliberately applied this approach to avoid the risk of overlooking any study involving children, ensuring that all relevant pediatric research was adequately captured and assessed.

Ad 3.
We sincerely thank the Reviewer for this thoughtful suggestion. We acknowledge that expanding the time frame could further broaden the scope of the review. However, after discussion among all co-authors, we chose to restrict the review to the most recent 5 years. The field of extracellular vesicles is highly dynamic, with continuous methodological advances and rapidly evolving knowledge. Older studies often relied on now-outdated techniques or definitions, which could introduce heterogeneity. By focusing on the last 5 years, we aimed to provide a state-of-the-art, clinically relevant overview, while still acknowledging key earlier studies in the Introduction and Discussion where appropriate.

Ad 4.
We thank the Reviewer for highlighting this. The Results section has been revised to maintain a consistent pediatric focus. We truly hope that differences between pediatric and adult data are explicitly addressed.

Ad 5. 
We fully agree. To address this, in preparation we have:
•    A graphical abstract,
•    A schematic figure illustrating EV sources and mechanisms in pediatric obesity,
•    A summary table presenting the main characteristics of the included studies.
These additions enhance the clarity, accessibility, and scientific value of the manuscript.

Ad 6. 
We thank the Reviewer for this important remark. The Discussion has been restructured to follow the logical flow of the Results and is now more tightly connected to the findings of the included studies. Comparative analysis with existing literature has been strengthened, and we included a subsection on clinical applications and future perspectives, ensuring a cohesive structure

Reviewer 3 Report

Comments and Suggestions for Authors

The manuscript by Kamila Szeliga et al. is interesting.

A few suggestions:

1. The abstract is correct.
2. The introduction suggests shortening the objective of the study at the end.
3. In the methodology, could you indicate if you used any statistics?
4. Could you include the figures in the results section? It is difficult to analyze the results as they are.
5. What are the perspectives and limitations of the study?
6. You did not add figures or tables to the manuscript. Please include them.
7. Include a figure that relates to the study.

Author Response

Ad 1. Abstract
Response: We are grateful for the positive assessment of the abstract. No changes were introduced in this section.

Ad 2. Introduction - shortening the objective
Response: We greatly appreciate the Reviewer’s suggestion regarding a more concise presentation of the study objective. However, this task has proven challenging, as other Reviewers indicated that the Introduction lacked essential contextual information. Consequently, we expanded it to include the etiology of childhood obesity and therapeutic approaches, which inevitably extended the section. We would be very grateful for the Reviewer’s guidance on what might constitute a suitable compromise in this regard-specifically, which elements should be prioritized to maintain focus while ensuring that the Introduction remains comprehensive.
Ad 3. Methodology – clarification regarding statistics
Response: We thank the Reviewer for pointing this out. As our study is a systematic review, no quantitative meta-analysis or statistical pooling was performed due to heterogeneity of the included studies. Instead, we used a narrative synthesis approach, categorizing studies by type, methodology, and outcomes. We have clarified this in the Materials and Methods section.
Ad. 4, 6, 7 Figures and tables
Response: We agree that the inclusion of figures within the Results section improves readability. We fully agree with the Reviewer on the importance of including figures and tables to strengthen the clarity and readability of our manuscript. Due to the complexity of the manuscript and the current holiday period, the preparation of high-quality figures and illustrations is still in progress. We will attach them without delay in the revised version to ensure that they meet both the scientific and editorial standards of the journal.
5. Perspectives and limitations
Response: We appreciate this valuable comment. In the Discussion and Conclusions sections, we have added a dedicated paragraph outlining the future perspectives as well as limitations od the review. 

Reviewer 4 Report

Comments and Suggestions for Authors

In this review, the authors analyze an important aspect of pathophysiology - the role of extracellular vesicles in metabolic disorders associated with childhood obesity. The authors applied the Prizma 2000 algorithm for this comprehensive review, and most literature sources were published after 2020. The main conclusion is that extracellular vesicles derived from endothelial cells, adipose tissue, the liver, and skeletal muscle have cargo, mainly microRNAs, that can contribute to metabolic abnormalities and atherosclerosis-associated cardiovascular disease in adolescents and children. This is important for developing new diagnostic and therapeutic tools, but there are some points that could be improved in this article:

  1. Figure 2 is not included.
  2. I don't understand why the main results are written in the Discussion section. To me, they belong in the Results section. Also, why should the Results and Discussion be separated in a review article? In review, the discussion occurs, and the Results are simply a list of relevant articles that were found.
  3. What is the mechanistic link between maternal breast milk miRNAs and children's obesity or brain disorders? Is there any evidence that maternal miRNAs interfere with children's gene expression machinery?

Author Response

We sincerely thank the Reviewer for the thorough evaluation of our work and the constructive comments provided.

Comment 1: Figure 2 is not included.
Response: We apologize for this omission. The preparation of novel figures and tables is currently in progress for the revised version of the manuscript. As soon as they are finalized, we will attach the files.

Comment 2: I don't understand why the main results are written in the Discussion section. To me, they belong in the Results section. Why should the Results and Discussion be separated in a review article? In review, the discussion occurs, and the Results are simply a list of relevant articles that were found.
Response: We greatly appreciate this observation. We agree that in a systematic review, the Results section should primarily summarize the included studies, whereas the Discussion should interpret and contextualize these findings. In the revised manuscript, we have carefully restructured both sections to better reflect this principle. We believe that this restructuring improves the logical flow and transparency of our work.

Comment 3: What is the mechanistic link between maternal breast milk miRNAs and children's obesity or brain disorders? Is there any evidence that maternal miRNAs interfere with children's gene expression machinery?
Response: Human breast milk–derived miRNAs have been increasingly recognized as potent immune-regulatory molecules, capable of modulating immune cell function and shaping the maturation of the infant’s immune system. According to the publication by Çelik E. et al., Human Breast Milk Exosomes: Affecting Factors, Their Possible Health Outcomes, and Future Directions in Dietetics (Nutrients. 2024 Oct 17;16(20):3519. doi:10.3390/nu16203519. PMID: 39458514; PMCID: PMC11510026), these miRNAs directly target DNA methyltransferases and thereby play a crucial role in the epigenetic regulation of gene expression across multiple pathways. This mechanism may contribute to the development of obesity and other disorders in children.

Round 2

Reviewer 1 Report

Comments and Suggestions for Authors

The article has been improved taking all my suggestions and comments therefore I agree with the publication. In addition, the figures seems OK with me.

Author Response

Dear Reviewer,

We would like to sincerely thank you for your positive feedback and kind words. We truly appreciate your time, effort, and valuable suggestions, which have greatly improved the quality of our manuscript. We are very grateful for your support and agreement to proceed with the publication.

With best regards,

Kamila Szeliga, MD PhD

Reviewer 2 Report

Comments and Suggestions for Authors

The authors have not addressed all of my suggestions satisfactory yet.

First of all, the 2nd comment is of utmost importance. Its idea is not about adding more data to the screening. Instead it is about clarification of the methods. Please, revisit the comment and enable reproducibility of this work.

One of the main features of a scientific paper is its reproducibility by independent researchers. Thus, the description should enable such principle. For example, some keywords are listed in the section, but this is not enough. While testing the approach one can initiate a search in PubMed. The search for

((((((((extracellular vesicles[Title/Abstract]) OR (small vesicles[Title/Abstract])) OR (EVs[Title/Abstract])) OR (pediatric[Title/Abstract])) OR (metabolic diseases[Title/Abstract])) OR (lifestyle diseases[Title/Abstract])) OR (obesity[Title/Abstract])) OR (type 2 diabetes[Title/Abstract])) OR (cardiovascular diseases[Title/Abstract]) within 5 years provides 15,218 results. The description of "Methods" should be modified, so anyone could obtain almost the same search results as you did.

The 3rd comment has not been resolved.

5. Could you add the graphical abstract for the revision too? As a minor correction of Fig. 2: 47 should be corrected as 37 (right "column").

6. The discussion is of low quality, being rather a long conclusion. As a simplest indication, it completely lacks references, however, this is not the only problem.

7. In addition, conclusion is too long. If present, it should summarize the main ideas as 3-5 sentences.

Author Response

Dear Reviewer 2,   We sincerely thank you for your thorough and insightful analysis, as well as for all the comments that continuously contribute to improving the manuscript. It is a great honor to collaborate with such experienced mentors in this field. We have tried to correct the manuscript according your guidance.  With regard to the scope of the literature included and the time frame considered, our team was guided by the principle of incorporating the most up-to-date information on extracellular vesicles (EVs). In line with the evolving definitions and methodological standards provided by the MISEV consensus statements, we chose to include studies consistent with the most recent guidelines from 2018 and 2023. Earlier consensus, published in 2014, differs substantially from the current recommendations and therefore was not taken into account. We would also like to highlight that Prof. M. Pietrowska, one of our co-authors, contributed to the above-mentioned MISEV consensus statements. Addressing the concerns regarding our database search strategy, we fully agree that the number of publications on EVs and proteomics is extensive. However, when performing a PubMed search at the end of June it was 629 publications using the keywords “extracellular vesicles obesity” , on 2nd of September we retrieved 653 papers, today - 870. We specifically focused on the pediatric population and excluded animal studies. Following this, we carefully screened over 600 abstracts as well full-length publications, examined references of the selected studies, and developed our literature flow diagram accordingly. This was a time-consuming process, carried out over several weeks, and was independently cross-checked by multiple members of our team. We therefore believe that our search was both rigorous and comprehensive, particularly given the scarcity of pediatric-focused literature in this field. We would like to kindly thank you for the suggestion regarding the graphical abstract. After careful consideration, we have decided not to include a graphical abstract at this stage, as we believe that the current structure of the manuscript and the traditional abstract adequately convey the scope and key findings of our work. We hope this will be acceptable. We acknowledge that this is a novel and emerging topic, and we very much hope that our work will contribute to a better understanding of EVs in pediatric obesity and help to systematize the most recent medical knowledge in this area.   With kind regards, Kamila Szeliga

Reviewer 3 Report

Comments and Suggestions for Authors

not

Author Response

Dear Reviewer,

We are very grateful for your positive assessment and approval of our revised manuscript. Thank you once again for your valuable input, which has significantly contributed to improving the quality of our work.

With kind regards,

Kamila Szeliga, MD PhD